# Humoral Response Induced by Prime-Boost Vaccination with the ChAdOx1 nCoV-19 and mRNA BNT162b2 Vaccines in a Teriflunomide-Treated Multiple Sclerosis Patient

**DOI:** 10.3390/vaccines9101140

**Published:** 2021-10-06

**Authors:** Yves Michiels, Nadhira Houhou-Fidouh, Gilles Collin, Jérôme Berger, Evelyne Kohli

**Affiliations:** 1Community Pharmacy, Center for Primary Care and Public Health (Unisanté) Lausanne 1011, University of Lausanne, 1015 Lausanne, Switzerland; jerome.berger@unisante.ch; 2Pharmacie Michiels, Research Department, 21600 Longvic, France; 3Laboratoire de Virologie, Hôpital Bichat, Sorbonne Paris Cité, Université Paris Diderot, AP-HP, 75018 Paris, France; nadira.houhou@aphp.fr (N.H.-F.); gilles.collin@aphp.fr (G.C.); 4School of Pharmaceutical Sciences, University of Geneva, 1205 Geneva, Switzerland; 5Institute of Pharmaceutical Sciences of Western Switzerland, University of Lausanne, 1015 Lausanne, Switzerland; 6UMR INSERM/uB/AGROSUP 1231, Team 3 HSP-Pathies, Labellisée Ligue Nationale Contre le Cancer and Laboratoire d’Excellence LipSTIC, 21000 Dijon, France; evelyne.kohli@u-bourgogne.fr; 7UFR des Sciences de Santé, Université de Bourgogne, 21000 Dijon, France; 8Univesity Hospital, 21000 Dijon, France

**Keywords:** SARS-CoV-2, BNT162b2 vaccine, ChAdOx1 nCoV-19 vaccine, antibody response, teriflunomide

## Abstract

Patients with multiple sclerosis (MS) are treated with drugs that may impact immune responses to SARS-CoV-2 vaccination. Evaluation of “prime-boost” (heterologous) vaccination regimens including a first administration of a viral vector-based vaccine and a second one of an mRNA-based vaccine in such patients has not yet been completed. Here, we present the anti-spike protein S humoral response, including the neutralizing antibody response, in a 54-year-old MS patient who had been treated with teriflunomide for the past 2 years and who received a heterologous ChAdOx1 nCoV-19/ BNT162b2 vaccination regimen. The results showed a very strong anti-S IgG response and a good neutralizing antibody response. These results show that teriflunomide did not prevent the development of a satisfactory humoral response in this MS patient after vaccination with a ChAdOx1 nCoV-19/ BNT162b2 prime-boost protocol.

## 1. Introduction

Vaccination against severe acute respiratory syndrome coronavirus 2 (SARS-CoV-2) is recommended for all patients with multiple sclerosis (MS) [1]. However, MS-disease modifying therapies (MS-DMTs), which act on the immune system and modify humoral and cellular immune responses, can have an impact on vaccine responses [2]. A literature review published in September 2020, which focused on humoral responses induced by vaccines other than those against SARS-CoV-2 infection, showed that vaccine responses among MS patients receiving DMT varied depending on the vaccine, the situation (booster or new antigen) and the type of MS treatment [3]. Indeed, vaccine responses were reported to be reduced by varying degrees in patients treated with glatiramer acetate, teriflunomide, fingolimod or natalizumab, and a significant reduction in response was observed in patients under anti-CD20 treatment (Ocrelizumab, Rituximab), in particular towards new antigens. However, few studies providing quantitative and qualitative evaluations of humoral responses in patients treated with MS-DMT have been published so far. A first case of vaccine failure in an MS patient receiving MS-DMT has been reported, with a patient on ocrelizumab showing an absence of postvaccination seroconversion and the development of a SARS-CoV-2 infection 19 days after a second dose of the mRNA BNT162b2 vaccine [4]. Two studies evaluating humoral responses to vaccination with a mRNA BNT162b2 vaccine or ChAdOx1 nCoV-19 in MS patients, either treated with MS-DMT or untreated, found that the titer of anti-SARS-CoV-2 spike protein IgG was lower in anti-CD20 and S1PR modulator compared to untreated patients or patients receiving other MS-DMT [5,6]. No specific data on teriflunomide were reported in these studies.

The postvaccination thrombosis has occurred in some patients aged < 55 years who had received one of the adenoviral vector SARS-CoV-2 vaccines. This led French authorities to modify the vaccination regimen in people aged < 55 years [7]. For those who had received a first dose of the ChAdOx1 nCoV-19 vaccine, the second dose of the ChAdOx1 nCoV-19 vaccine should be replaced by a dose of an mRNA vaccine, such as the BNT162b2 vaccine, while respecting the planned time interval between doses [8]. Evaluation of this “prime-boost” or heterologous vaccination regimen, has not yet been completed [9].

Here, we present an evaluation of the humoral response to SARS-CoV-2 vaccination, including assessment of the neutralizing antibody response, in a 54-year-old MS patient who had been treated with teriflunomide for the past 2 years and who received a heterologous vaccination regimen.

## 2. Case Description

The patient was a 54-year-old woman who began treatment with glatiramer acetate in 2003 and then switched treatment with teriflunomide (1 tablet/day) in July 2019. Prescription renewal data indicate that the patient showed good medication adherence. She received a first-dose SARS-CoV-2 vaccine (ChAdOx1 nCoV-19) on 13 March 2021 and a second-dose SARS-CoV-2 vaccine (mRNA BNT162b2) on 12 May 2021, i.e., almost 9 weeks after the first dose.

Humoral response induced by vaccination with a heterologous protocol (ChAdOx1 nCoV-19 /mRNA BNT162b2). Two sets of four serological tests were performed: the LIAISON^®^ SARS-CoV-2 Trimeric S IgG assay (DiaSorin, Saluggia, Italy) and the Architect^®^ anti-spike test (Abbott, Rungis, France), both directed against the SARS-CoV-2 S-protein; the iFlash^®^-2019-nCoV NAb (Orgentec^®^, Trappes, France) assay to measure SARS-CoV-2 neutralization antibody levels, and the Architect^®^ anti-N SARS-CoV-2 IgG assay (Abbott, Rungis, France) to assess a potential previous natural infection [10,11,12]. The first set of tests was performed 14 and 28 days after administration of the second dose of the vaccine (Table 1). The results showed a strong anti-S antibody response (>40,000 and >40,000 AU/mL at day14 and 28 post second vaccination, respectively) and a good neutralizing antibody response. 

## 3. Discussion

Both types of vaccine received by the patient described in our report encode the same antigen, i.e., the S protein of the first SARS-CoV-2 strain. The ChAdOx1 nCoV-19 vaccine is a non replicative recombinant viral-vectored vaccine; the replication-deficient simian adenovirus vector (ChAdOx1) contains the full-length S protein expressed in its trimeric prefusion conformation [13,14]. The mRNA BNT162b2 vaccine also encodes the full-length S protein, but in this vaccine the protein is stabilized in the prefusion conformation by proline substitutions at residues 986 and 987 [15]. Previous studies have shown that heterologous vaccination regimens, in which the same antigen (*Plasmodium falciparum* or *Influenza* A) is administered by different viral vectors in order to avoid the antivector memory response, can induce satisfactory immune responses [16,17]. Indeed, this strategy is currently being used for the Gam-COVID-Vac vaccine, where two different adenovirus serotypes (26 and 5) are being used in the prime and booster doses [18].However, the prime-boost regimen used in our patient differed significantly from that used for the Gam-COVID-Vac vaccine in that the boost in our case was achieved with an mRNA vaccine instead of a vaccine using another viral vector. Unexpectedly, the anti-S IgG response observed after the booster dose was very high (>40,000 AU/mL from day 14 after the second dose) and much greater than those we observed previously in nonimmune-suppressed subjects vaccinated with homologous vaccination regimens, including either two doses of the ChAdOx1 nCoV-19 or two doses of mRNA BNT162b2 vaccines (data not shown). Teriflunomide selectively and reversibly inhibits dihydroorotate dehydrogenase, a key enzyme in the de novo synthesis of pyrimidine and has a cytostatic effect on proliferating T- and B-lymphocytes. Teriflunomide may, therefore, have inhibited, at least partly, the anti-vector response in our patient, and thus may have allowed prolonged antigen expression. As reported by Geiben-Lynn et al. for rAd5 transgene expression in nude mice, this prolonged antigen expression would likely have been beneficial for B-cell priming [19]. The authors reported that vaccine antigen clearance is mostly dependent on the strength of adaptive immune responses, and that these responses are associated with the strength of the elicited memory cellular responses, with T lymphocytes playing a central role in damping rAd5 transgene expression. Thus, prolonged expression of the Sprotein after the priming dose may have compensated for the immunosuppressive effect of teriflunomide on the anti-S T cell immune response, and consequently on the memory B cell response. Using an mRNA-based vaccine for the booster dose in SARS-CoV-2 regimens may therefore allow the problem of neutralizing anti-vector antibodies to be avoided, potentially making the vaccines more effective. Another unexpected finding from our study was that the strong total anti-S antibody response was not associated with a strong neutralizing antibody titer, as it was similar to that observed in our experience for the non immunosuppressed subjects mentioned above. These results highlight the complexity of evaluating qualitative humoral responses at the individual level. However, the very high titer of anti-S IgG observed in our MS patient, together with the neutralizing antibody titer similar to that observed in subjects without immunosuppressive therapy after two doses of ChAdOx1 nCoV-19 or mRNA BNT162b2 vaccines, show that teriflunomide may not prevent a quality humoral anti-SARS-CoV-2 response. As the neutralizing antibodies’ titer during the peri-infection period has been recently correlated with protection, it can be assumed that our patient is as protected as the non immunosuppressed subjects who received homologous protocols [20]. 

## 4. Conclusions

Although the case presented did not allow any correlations to be made between serological findings and the level of protection afforded by SARS-CoV-2 vaccines in DMT-treated MS patients, the case presented here shows that teriflunomide did not prevent the development of a satisfactory humoral response in an MS patient who received a prime-boost ChAdOx1 nCoV-19/m RNA BNT162b2 heterologous vaccination regimen.

## Figures and Tables

**Table 1 vaccines-09-01140-t001:** Humoral response in a patient treated with teriflunomide and vaccinated with a heterologous protocol (ChAdOx1 nCoV-19-mRNA BNT162b2).

Test	Antibody Titers (AU/mL) after a Heterologous Vaccination Regimen: ChAdOx1 nCoV-19-mRNA BNT162b2	
	D14 Post-2nd Vaccination	D28 Post-2nd Vaccination
LIAISON^®^ SARS-CoV-2 TrimericS IgG assay (DiaSorin^®^; threshold: 13 AU/mL) [10]	>800	>800
Architect^®^ anti-spike test (Abbott; threshold: 50 AU/mL) [11]	>40,000	>40,000
iFlash-2019-nCoV Nab neutralization test^®^ (Orgentec; threshold: 10 AU/mL) [12]	600	612
Architect^®^ anti-N SARS-CoV-2 IgG assay (Abbott)	Negative	Negative

AU: arbitrary unit.

## Data Availability

Data available on request due to restrictions, e.g., privacy or ethics.

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
