# Peer review of "Humoral Response Induced by Prime-Boost Vaccination with the ChAdOx1 nCoV-19 and mRNA BNT162b2 Vaccines in a Teriflunomide-Treated Multiple Sclerosis Patient"

_vaccines, 2021, doi:10.3390/vaccines9101140_

Round 1

Reviewer 1 Report

Despite of some interest, there are several points that need to be addressed in this paper. first of all, since it is a case report, the conclusions cannot be extended to whole population undergoing TLFN treatment and should be carefully considered. Also, the case used as a comparison is not correct. It is known that a differemt immune response occurs when different vaccines are administred, and this is also the case of SARS-CoV-2 vaccine in which heterologous vaccination may even boost the immune response. Confirmatory data on this are awaited. Whatever the case, the two cases cannot be compared.

Author Response

Our answer:

- we agree on the fact that even if the antibody responses were analyzed with the same method and in the same laboratory, it is difficult to compare the antibody levels for these two patients. However, results from our experience for non-immunosuppressed subjects are interesting for us to estimate the level of the humoral response in our patient. Thus, we propose to remove patient 2 (homologous protocol with 2 doses of ChAdOx1 nCoV-19) from the results section to avoid a direct comparison with our case and only mention it (as data not shown) in the discussion together with another non-immunosuppressed subject who received a homologous protocol with 2 doses of the mRNA BNT162b2 vaccine.

-we also agree on the first comment and modified the end of the discussion as follows, the conclusion reporting only on our patient:

These results highlight the complexity of evaluating qualitative humoral responses at the individual level. However, the very high titer of anti-S IgG observed in our MS patient, together with the neutralizing antibody titer similar to that observed in subjects without immunosuppressive therapy after two doses of ChAdOx1 nCoV-19 or mRNA BNT162b2 vaccines, show that teriflunomide may not prevent a quality humoral anti-SARS-CoV-2 response”.

Reviewer 2 Report

This report is important for social needs  and valuable data. Unfortunately, the patient is only 2 persons, so it is hard to agree to author`s conclusion.  

Author Response

Our answer: we agree with this comment. The conclusion reporting only on our patient, we modified the end of the discussion (see reviewer 1). We are aware that our study includes only one patient, but the objective of this case report is to bring some information to the practitioners who care for patients with multiple sclerosis waiting for publications including a high number of patients.

Reviewer 3 Report

The authors present an interesting "case report" on the efficacy of heterologous anti-SARS-Cov-2 vaccination in a patient with multiple sclerosis treated with teriflunomide.
In general, the manuscript is well written, the introduction is very clear and complete, the case and the control patient are well presented, the methods are well described, the results are interesting and the discussion is adequate, balanced and above all not high-sounding.
I believe the manuscript deserves to be published.
I have only a few minor observations in an undefined order of impordance:
1) Abstract section: I believe that the term "anti-S humoral response" should be better explained;
2) The results section should be separate from the patient description and should have a title and a progressive number (3?)
3) third to last line of the discussion: I assume that the full stop after the bibliographic citation [20] is a typo ...

Author Response

Our answer
I have only a few minor observations in an undefined order of impordance:

Our answer
1) Abstract section: I believe that the term "anti-S humoral response" should be better explained;

We have now corrected the abstract as follows: the anti- spike glycoprotein S humoral response”

2) The results section should be separate from the patient description and should have a title and a progressive number (3?) : This has been done

3) third to last line of the discussion: I assume that the full stop after the bibliographic citation [20] is a typo ... yes, it is, this has now been corrected.

Round 2

Reviewer 1 Report

the authors responded to the comments properly.

Reviewer 2 Report

I understand the situation that the objective of this case report is to bring some information to the practitioners who care for patients with multiple sclerosis.